# HGSMDA: miRNA–Disease Association Prediction Based on HyperGCN and Sørensen-Dice Loss

**DOI:** 10.3390/ncrna10010009

**Published:** 2024-01-26

**Authors:** Zhenghua Chang, Rong Zhu, Jinxing Liu, Junliang Shang, Lingyun Dai

**Affiliations:** School of Computer Science, Qufu Normal University, Rizhao 276826, China; changzhenghua313@126.com (Z.C.); sdcavell@126.com (J.L.); shangjunliang110@163.com (J.S.); dailingyun_1@163.com (L.D.)

**Keywords:** miRNA–disease association, HyperGCN, Sørensen-Dice loss, prediction

## Abstract

Biological research has demonstrated the significance of identifying miRNA–disease associations in the context of disease prevention, diagnosis, and treatment. However, the utilization of experimental approaches involving biological subjects to infer these associations is both costly and inefficient. Consequently, there is a pressing need to devise novel approaches that offer enhanced accuracy and effectiveness. Presently, the predominant methods employed for predicting disease associations rely on Graph Convolutional Network (GCN) techniques. However, the Graph Convolutional Network algorithm, which is locally aggregated, solely incorporates information from the immediate neighboring nodes of a given node at each layer. Consequently, GCN cannot simultaneously aggregate information from multiple nodes. This constraint significantly impacts the predictive efficacy of the model. To tackle this problem, we propose a novel approach, based on HyperGCN and Sørensen-Dice loss (HGSMDA), for predicting associations between miRNAs and diseases. In the initial phase, we developed multiple networks to represent the similarity between miRNAs and diseases and employed GCNs to extract information from diverse perspectives. Subsequently, we draw into HyperGCN to construct a miRNA–disease heteromorphic hypergraph using hypernodes and train GCN on the graph to aggregate information. Finally, we utilized the Sørensen-Dice loss function to evaluate the degree of similarity between the predicted outcomes and the ground truth values, thereby enabling the prediction of associations between miRNAs and diseases. In order to assess the soundness of our methodology, an extensive series of experiments was conducted employing the Human MicroRNA Disease Database (HMDD v3.2) as the dataset. The experimental outcomes unequivocally indicate that HGSMDA exhibits remarkable efficacy when compared to alternative methodologies. Furthermore, the predictive capacity of HGSMDA was corroborated through a case study focused on colon cancer. These findings strongly imply that HGSMDA represents a dependable and valid framework, thereby offering a novel avenue for investigating the intricate association between miRNAs and diseases.

## 1. Introduction

MiRNAs (microRNAs) belong to a class of endogenous non-coding single-stranded RNAs. They play a critical role in regulating biological phenomena through mechanisms such as cleavage or translational repression [1]. The dysregulation of miRNA expression can lead to alterations in the levels of gene expression for their target genes, thereby playing a vital role in the initiation and progression of specific illnesses. Hence, uncovering the correlation between miRNAs and diseases is of utmost significance for medical researchers, as it enables them to enhance their comprehension of the intricate pathological mechanisms underlying various diseases. Given that traditional biological experiments to infer the correlation between miRNAs and diseases are costly and take up a significant amount of time, computer-based methods have emerged as a viable and efficient strategy to tackle this challenging task. In recent years, computational methods commonly employed for this purpose can be categorized into two groups: similarity-based predictive models and machine learning-based predictive models.

Fundamental predictive models, which rely on the assumption of similarity, propose that miRNAs sharing functional similarities are linked to diseases exhibiting comparable phenotypes, and the reverse is also true. These models are based on the principle that entities with similar characteristics or traits are inclined to have related functions or associations. In similarity-based prediction models, it is crucial to acquire similarity scores between various biomedical entities. Chen et al. [2] came up with a Randomized Wandering and Restart-based miRNA–disease Association (RWRMDA) method, which first constructs a functional similarity network of miRNAs associated with diseases. Zhong et al. [3] introduced a new method utilizing non-negative matrix decomposition to rank candidate disease miRNAs for prediction. To fully incorporate similarity information into miRNA–disease heteromorphic networks, MDHGI [4] developed a miRNA–disease heteromorphic map using matrix decomposition techniques. In the study by BNPMDA [5], a dichotomous network recommendation algorithm was used to represent the relationship between miRNAs and diseases as a dichotomous network, and the topology of this network and the attribute information of the nodes were utilized to predict possible correlations between miRNAs and diseases. In order to accomplish this, transfer weights were assigned to establish connections between resource allocation and miRNAs and diseases, while also considering bias ratings. These means have yielded excellent results in predicting miRNA–disease correlations, but these similarity-based models tend to be overly reliant on known associations between miRNAs and diseases.

Machine learning-based predictive models are also commonly utilized to discover associations between miRNAs and diseases according to specific criteria. Feature extraction is a vital stage in most machine learning models, as it significantly influences the forecast outcomes of the classifier. Xu et al. [6] proposed a means referred to as MTDN that inputs characteristics derived from the network into an SVM classification model and then feeds the features back into a predictive model. IMCMDA [7] is a new inductive matrix-completion approach that utilizes known associations and integrated miRNA similarities and disease similarities to complete the association of missing miRNAs with diseases. HDMP [8] predicts built upon weighting the k most similar neighbors, combining the informational value of disease terms and the phenotypic similarity between diseases. ABMDA [9] enhances the precision of a given learning algorithm by incorporating weak classifiers capable of scoring samples. Shang et al. [10] proposed a weighted bi-level network based miRNA–disease association (BLNIMDA) method, which calculates association scores based on bidirectional information distribution strategies and association types to ensure comprehensive and accurate predictions. Machine learning-based models, greatly improve efficiency, but these models require high quality of features, such as miRNA expression levels, biological functions, biological pathways involved, and network properties of miRNAs associated with known diseases.

In the past few years, graph convolutional networks (GCNs) have demonstrated strong performance capabilities in learning graphical representations. The method has garnered significant attention and has found extensive application in fundamental research areas, for example the prediction of drug-target interactions [11,12,13] and tumor-associated gene prediction [14]. Also, given the excellent performance of graph convolutional networks in link prediction tasks, in recent years it has begun to be applied to inferring links between miRNAs and diseases. NIMCGCN [15] employs GCN, which fully extracts the nonlinear characteristics of miRNAs and diseases and predicts associations by achieving feature enhancement through multichannel attention. MMGCN [16] utilizes a GCN encoder to extract features. It further employs a multi-channel attention strategy to assign weights to different features for predicting associations. GCNA-MDA [17] utilizes an autoencoder to extract representations, while GCN is used to capture topological information to forecast miRNA–disease correlations. GCN-based models have the capability to aggregate more comprehensive node information, thereby enhancing the predictive ability of association. However, GCN is a locally aggregated graph convolution algorithm that can only consider the information of a node’s first-order neighbors at each layer and cannot aggregate multiple nodes. If the number of layers of the GCN is increased to handle information from multiple nodes, then nodes in the GCN may begin to receive information from most or all other nodes in the graph. This may lead to an oversmoothing problem, where the feature representations of different nodes tend to be similar, making it difficult for the network to capture differences between nodes. Additionally, GCNs usually require the structure and feature information of the entire graph for forward propagation. This means that for very large graphs, GCN may be difficult to scale as it requires a lot of memory and computational resources.

This paper introduces a novel approach, named HGSMDA, for forecasting miRNA–disease associations. HGSMDA is built upon the foundation of HyperGCN and utilizes the Sørensen-Dice loss function. The contributions of the method are as follows:Information about miRNAs, information about diseases, and information about known miRNA–disease associations were integrated. By integrating this information into the HGSMDA model, we were able to more fully characterize the relationship between miRNAs and disease.HyperGCN was introduced to construct a miRNA–disease heterogeneous hypergraph using hypernodes, and GCNs were trained on the graph to aggregate information.The Sørensen-Dice loss function is employed to evaluate the likeness between the predicted outcomes and the actual values. This facilitates a more precise evaluation of the model’s capability.

## 2. Data and Experiments

### 2.1. Datasets

The miRNA–disease association data utilized in the present study were sourced from the HMDD v3.2 [18]. This database is one of the largest human microRNA disease databases in the world and collects human microRNA sequencing data from all over the world. A total of 12,446 associations between 853 miRNAs and 591 diseases were carefully selected for this study [19,20,21]. These associations have been verified through experimentation and are considered reliable. Next, we labeled the identified miRNA–disease correlations as positive samples, indicated by the number 1. For other samples, the absence of correlations between miRNA and disease is indicated by the number 0. Predicting associations between miRNAs and diseases is dependent on the hypothesis that miRNAs exhibiting analogous functions are more inclined to be linked with diseases that possess analogous characteristics. Therefore, understanding the relationships between disease–disease and miRNA–miRNA becomes crucial in this context. In this research investigation, we employed a comprehensive approach that considered multiple factors to determine the resemblance between miRNA–miRNA and disease–disease. These factors include similarity in miRNA function, resemblance in miRNA sequence, semantic resemblance in disease, target-based disease resemblance, and nuclear resemblance in the Gaussian interaction spectrum between miRNA and disease.

### 2.2. Parametric Analysis

In our training process, we established the batch size as 128, the dimension of the node features was established as 128, and the number of hypernodes was established as 64. In addition, to mitigate the issue of overfitting, we set the dropout to 0.5 to randomly ignore certain neurons. We used several assessment metrics to provide a comprehensive assessment of the forecast results, including the area under the curve (AUC) for receiver operating characteristics, the area under the precision/recall curve (AUPRC), precision, F1-score, and accuracy. To ensure a reliable evaluation of the predicted outcomes, we performed multiple replications of all experiments. This ensures the stability and dependability of the assessment results. The model’s predictive performance is influenced by hyperparameters. Hence, we employed Cross-validation with a 5-fold approach to analyze the effect of certain parameters on the predicted outcomes and to conduct a comparison of the evaluation metrics values.

#### 2.2.1. Dropout Parameter Settings

Dropout is critical for tuning deep learning models. During training, dropout uses a randomized approach to zero out the output of a portion of neurons. This reduces the model’s dependence on specific neurons and thus reduces the risk of overfitting. Figure 1 illustrates the evaluation metric scores for different dropouts. For optimal performance, we chose a suitable dropout of 0.5 after tuning.

#### 2.2.2. Feature Embedding Dimension

The dimension of the learnable parameter matrix associated with a feature is influenced by its embedding size. The evaluation metric scores for different feature dimensions, specifically {32, 64, 128, 256}, are depicted in Figure 2. We can see that there are significant differences in precision between different feature dimensions. We chose a suitable feature dimension of 128 for optimal performance after tuning.

#### 2.2.3. The Number of Hypernodes

The structure of the hypergraph is influenced by the number of hypernodes. Hence, we conducted further analysis to examine the impact of the quantity of hypernodes on the prediction outcomes. Figure 3 displays the evaluation metric scores for different numbers of hypernodes, namely {8, 16, 32, 64, 128}. It is important to note that having too many hypernodes can hinder the extraction of useful information and potentially complicate the graph structure due to the random initialization of hypernode representations. Consequently, we have determined that the optimal number of hypernodes is 64.

## 3. Results and Discussion

### 3.1. Comparison with Other Methods

We conducted a comparison between our model and eight other methods on HMDD v3.2, namely NIMCGCN [15], MMGCN [16], ERMDA [22], HGANMDA [23], AGAEMD [24], MINIMDA [25], MAGCN [26] and AMHMDA [27]. For all comparison methods, the initial similarity data used is the one already available for each respective method. Furthermore, we maintained the settings, parameters, and the portion that dynamically acquires specific similarities for the other methods.

NIMCGCN [15] uses GCN to learn miRNA and disease potential characteristic representations. It inputs the learned characteristics into the NIMC model to produce a matrix of association complements to forecast miRNA–disease associations.MMGCN [16] uses a GCN encoder to obtain characteristics under different similarity views separately and forecasts miRNA–disease associations by utilizing multi-channel attention to enhance the learned potential representation of association prediction.ERMDA [22] proposed a resampling strategy to construct multiple subsets and applied feature selection methods to increase the diversity among these subsets. It then uses soft voting to forecast the connections of miRNAs with diseases.HGANMDA [23] constructs a heterogeneous graph, applies node-level attention to learn neighboring nodes, applies semantic-level attention to learn meta-paths, and lastly employs a bilinear decoder to reconstruct miRNA–disease associations.AGAEMD [24] creates heterogeneous matrices and uses autoencoders in miRNA–disease networks to polymerize information and reconstruct miRNA–disease association networks.MINIMDA [25] constructs disease similarity networks to obtain embedding representations by mixing higher-order neighborhood information, which is fed into a multilayer perceptron (MLP) to forecast potential connections between miRNAs and diseases.MAGCN [26] used lncRNA-miRNA interactions to predict novel miRNA–disease correlations through graph convolutional networks with attentional mechanisms and convolutional neural network combiners.AMHMDA [27] constructed multiple similarity networks, introduced virtual nodes to construct heterogeneous hypergraphs, and used the output of graph convolutional networks to predict associations.

Figure 4 displays the results of multiple experiments for comparative analysis. HGSMDA had an AUC of 94.81%, AUPRC of 94.29%, ACC of 88.32%, and a recall of 87.36%. HGSMDA shows good performance on HMDD v3.2 compared to alternative approaches.

### 3.2. Ablation Experiments

To evaluate the significance of each module in HGSMDA, we created two different versions of the model, namely HGSMDA-L and HGSMDA-H, for comparative analysis. Specifically, to assess the impact of the Sørensen-Dice loss function on model performance, HGSMDA-L replaces the Sørensen-Dice loss function in the original modeling framework with the BCE loss function frequently used in classification tasks. To assess the impact of HyperGCN on model performance, HGSMDA-H replaces the miRNA–disease heteromorphic hypergraph with a miRNA–disease heteromorphic graph, and the model loses the ability to model hyperedges, thus no longer constituting HyperGCN.

Figure 5 shows a comparison of the evaluation metrics for HGSMDA-L, HGSMDA-H, and HGSMDA. We can find that the loss function Sørensen-Dice has a positive effect on the model predictions. This shows that the Sørensen-Dice loss function improves the performance of the model by operating on the intersection and union of the predictions with the true labels. The introduction of HyperGCN contributes to the prediction results of the model. This suggests that by utilizing hyperedges to capture higher-order relationships between nodes, HyperGCN is better able to capture the correlation and similarity between nodes, thus improving the performance of the model.

### 3.3. Case Study

Colon cancer, a malignant neoplasm affecting the gastrointestinal tract within the colon region, can be effectively detected at an early stage through the utilization of colonoscopy [28]. The global incidence of cancer has been progressively rising in recent times [29]. Consequently, the identification of prognostic and predictive biomarkers linked to colon cancer holds immense importance in the realm of colon cancer treatment. The investigation of miRNAs has garnered significant attention due to their involvement in colon cancer cell proliferation and resistance to therapeutic agents. For example, reduced let-7a expression is closely associated with colon carcinogenesis [30]. High expression of the gene encoding mir-21 is associated with low expression of the gene encoding the tumor suppressor protein PDCD4 [31]. The known miRNA–disease associations in the HMDD v3.2 dataset were employed as a training dataset. Subsequently, the HGSMDA model was employed to predict potential miRNAs for colon cancer. Following this, we screened the top 10 predicted miRNAs and validated these predicted associations based on the authoritative miRNA–disease association dataset dbDEMC [32]. The results are shown in Figure 6, and the obtained results were validated in the database. This comprehensive demonstration underscores the effectiveness and dependability of HGSMDA in accurately predicting miRNA–disease associations.

## 4. Methodology

### 4.1. HGSMDA Framework

Inspired by the AMHMDA [27] research, we introduced a new approach called HGSMDA to predict the association of miRNAs with diseases. HGSMDA is built upon the foundation of HyperGCN and utilizes the Sørensen-Dice loss function, as described in Figure 7.

The HGSMDA process is comprised of three primary stages:Extracting features: We constructed multiple miRNA and disease similarity networks and used GCN for extracting information from different perspectives.HyperGCN: We introduce HyperGCN to construct a miRNA–disease heteromorphic hypergraph using hypernodes, and train GCN on the graph to aggregate information.Measuring the degree of similarity: We leverage the attention mechanism to fuse the output of the HyperGCN layer in combination with a CNN (Convolutional Neural Network) for classification. We then use the Sørensen-Dice loss function to scale the degree of similarity between the predictions and the true values.

### 4.2. Extraction of Features

To investigate similarity data that aids in making association predictions, we constructed three different miRNA–miRNA networks and three different disease–disease networks. The collections of adjacency matrix for miRNAs and diseases are denoted as stated below:(1)$$B_{m}=\left\{B_{mf}\left[H_{mf}\right],B_{ms}\left[H_{ms}\right],B_{mg}\left[H_{mg}\right]\right\}$$
(2)$$B_{d}=\left\{B_{ds}\left[H_{ds}\right],B_{dt}\left[H_{dt}\right],B_{dg}\left[H_{dg}\right]\right\},$$
where $B_{mf}$ denotes the adjacency matrix of miRNA functional similarity matrices, $B_{ms}$ denotes the adjacency matrix of miRNA sequence similarity matrices, and $B_{mg}$ denotes the adjacency matrix of Gaussian interaction profile kernel similarity matrices. Similarly, where $B_{ds}$ denotes the adjacency matrix of the disease semantic similarity matrices, $B_{dt}$ denotes the adjacency matrix of target-based disease similarity matrices, and $B_{dg}$ denotes the adjacency matrix of Gaussian interaction profile kernel similarity matrices.

After constructing the interaction network, the GCN extracts the information in the miRNA and disease networks by iteratively propagating the information of the nodes and updating the representation of the nodes. The initial embedding of the GCN is achieved by generating random feature vectors with specified dimensions for each node. On this basis, the adjacency matrix of miRNA–miRNA and disease–disease interaction network was normalized:(3)$$J_{m}={\tilde{P}}_{m}^{-\frac{1}{2}}{\tilde{B}}_{m}{\tilde{P}}_{m}^{-\frac{1}{2}}$$
(4)$$J_{d}={\tilde{P}}_{d}^{-\frac{1}{2}}{\tilde{B}}_{d}{\tilde{P}}_{d}^{-\frac{1}{2}},$$
where $\tilde{B}$ denotes the adjacency matrix with the unit matrix added, and $\tilde{P}$ is the degree matrix of $\tilde{B}$.

### 4.3. HyperGCN

After obtaining an embedded representation that aggregates various similarity information of miRNAs and diseases, we construct a hypergraph of miRNAs and diseases through hypernodes. Within the hypergraph, hypernodes can connect to multiple nodes or other hypernodes. These hypernodes are the connection source of all miRNA nodes and disease nodes, which provides a new idea for miRNA–disease node research. The introduction of hypernodes allows us to incorporate unknown miRNA–disease associations into the hypergraph, thus exploring potential miRNA–disease associations more comprehensively. Furthermore, hypernodes have the ability to autonomously acquire node representations within the network, thus aggregating richer information in the hypergraph and improving the quality of connections between miRNAs and diseases. This approach avoids the use of formulas to directly calculate the association score, but instead computes the cosine similarity to obtain the adjacency matrix of the hypergraph, which better describes the association relationship between miRNAs and diseases, and can be described as:(5)$$C_{M_{i},v_{k}}=\frac{M_{i}\cdot v_{k}}{{\left\|M_{i}\right\|}_{2}{\left\|v_{k}\right\|}_{2}}$$
(6)$$C_{D_{j},v_{k}}=\frac{D_{j}\cdot v_{k}}{{\left\|D_{j}\right\|}_{2}{\left\|v_{k}\right\|}_{2}},$$
where $M_{i}$ represents the embedding of a previously obtained miRNA node i and $D_{j}$ represents the embedding of a previously obtained disease node j. v is a hypernode whose node characteristics are randomly initialized. Thus, for miRNA i-disease j pairs, the correlation between them can be quantified using variables $C_{M_{i},v_{k}}$ and $C_{D_{j},v_{k}}$.

We performed GCN on the graph. In the neural information propagation framework, we denote the information update of hypernode v as:(7)$$Z_{v}^{\left(\tau +1\right)}=\sigma \left({\left(\Theta^{\left(\tau \right)}\right)}^{T}\sum_{u\in N\left(v\right)}\left({\left[{\bar{A}}_{S}^{\left(\tau \right)}\right]}_{v,u}\cdot h_{u}^{\left(\tau \right)}\right)\right),$$
where τ is the epoch, $Z_{v}^{\left(\tau +1\right)}$ is the new hidden layer node representation of the node v, and $N\left(v\right)$ is the neighborhood of v, ${\left[{\bar{A}}_{S}^{\left(\tau \right)}\right]}_{v,u}$ is the weight of the regularized edge $\left\{v,u\right\}$, and $h_{u}^{\left(\tau \right)}$ is the implicit representation of the neighboring nodes in the previous stage. We employ the conventional graph convolution operation on the variable v, taking into consideration only the simple edges that are incident to it. For each node, its features are weighted and averaged according to the features of its neighboring nodes to update the node’s representation. We perform operations on each super node v∈V during each training period τ until convergence is achieved.

### 4.4. Measuring Similarity

We believe that the information of nodes in different layers contributes differently to the prediction results, after acquiring the representations of nodes from various layers, we utilize the layer-level attention mechanism to acquire the node representations at different layers, each assigned with varying degrees of significance. The layer-level attention mechanism assigns different importance to node representations in different layers by calculating the weights of node representations in each layer. This allows the model to pay more attention to node representations that have a greater impact on the prediction of miRNA–disease associations. In this way, the layer-level attention mechanism contributes to the final node representation, enabling the model to better capture information from different layers of node representations:(8)$${\hat{W}}_{i}=cnn\left(att\left(W_{i}^{c}\right)\right)$$
(9)$${\hat{F}}_{j}=cnn\left(att\left(F_{j}^{c}\right)\right),$$
where ${\hat{W}}_{i}$ indicates the ultimate embedding of miRNA, where ${\hat{F}}_{j}$ indicates the ultimate embedding of disease. Afterward, element-level multiplication was carried out on the embeddings of miRNA nodes and disease nodes. After that, we utilize a Feedforward Neural Network (FNN) to forecast the likelihood of association between miRNA–disease pairs.

To enhance the model’s performance, we utilized the Sørensen-Dice loss function to compute the loss during the training of the model:(10)$$L=1-\frac{X+\varepsilon }{U-X+\varepsilon },$$
where X is acquired:(11)$$X=\sum_{1}^{N}t_{i}q_{i},$$
where U is obtained by:(12)$$U=\sum_{1}^{N}\left(t_{i}^{2}+q_{i}^{2}\right),$$
where $q_{i}$ is the network prediction value, which is the value obtained after sigmoid and takes the value between (0, 1). $t_{i}$ is the target value, which can only be either 0 or 1.

ε is a smoothing factor that serves two purposes. First, it prevents the denominator from predicting zero. Generally, the output of the segmentation network goes through sigmoid or softmax, and there is no case where the output is absolutely 0. The smoothing factor is added here to prevent some extreme cases where the number of output bits is too small and causes the compiler to lose digits. Second, smoothing coefficients can operate to smooth out losses and gradients.

The Sørensen-Dice loss function pays more attention to the overlap of the predictions when calculating the loss. It works on the principle that when calculating the Sørensen-Dice loss function, the intersection and union of the predictions with the true labels need to be calculated in order to assess the degree of overlap of the predictions and consequently enhance the precision and resilience of the model. In binary classification problems, the Sørensen-Dice loss function focuses directly on the identification of positive samples, which is particularly important for miRNA–disease association prediction, as researchers are often more concerned with accurately identifying associations that are present than absent associations. In addition, the application of the Sørensen-Dice loss function reduces the loss value. We take the commonly used loss function Binary Cross Entropy Loss (BCE Loss) and compare it with the Sørensen-Dice loss function by taking the loss values after the first 100 batches as shown in Figure 8.

As seen in Figure 8, the loss value of the Sørensen-Dice loss function is lower than that of the BCE loss function. This is because the Sørensen-Dice loss function is much stricter. It takes into account not only their intersection but also their concatenation when calculating the similarity of two sets. Therefore, the Sørensen-Dice loss function requires more accuracy in the model’s prediction results and is more capable of reducing the disparity between the predicted consequences and the true consequences compared to the BCE loss function.

## 5. Conclusions

Improving the prediction of the relationship between diseases and miRNAs can greatly enhance human research on the pathogenesis of diseases. Presently, the predominant methods employed for predicting disease associations rely on Graph Convolutional Network (GCN) techniques. However, the Graph Convolutional Network algorithm, which is locally aggregated, solely incorporates information from the immediate neighboring nodes of a given node at each layer. Consequently, GCN cannot simultaneously aggregate information from multiple nodes. This constraint significantly impacts the predictive efficacy of the model. To tackle this problem, we propose a novel approach, based on HyperGCN and Sørensen-Dice loss (HGSMDA), for predicting associations between miRNAs and diseases. In the initial phase, we developed multiple networks to represent the similarity between miRNAs and diseases and employed GCNs to extract information from diverse perspectives. Subsequently, we draw into HyperGCN to construct a miRNA–disease heteromorphic hypergraph using hypernodes and train GCN on the graph to aggregate information. Finally, we utilized the Sørensen-Dice loss function to evaluate the degree of similarity between the predicted outcomes and the actual values, thereby enabling the prediction of associations between miRNAs and diseases. To comprehensively and objectively evaluate the predictive performance of the HGSMDA model, we employed methods such as five-fold cross-validation and case analysis to assess the model from different perspectives. The evaluation results indicate that HGSMDA exhibits excellent performance and can be used for miRNA–disease association prediction.

However, HGSMDA still has potential for further improvement. Above all, calculating miRNA and disease similarity scores using available information is challenging. For example, similarity between miRNAs can be assessed based on aspects such as their functions, sequences, or interaction networks, while similarity between diseases can be assessed based on aspects such as their phenotypes, gene expression profiles, or clinical features. Therefore, how to effectively integrate this multi-source information and translate it into comparable similarity scores is a complex issue. Second, our method may face computational efficiency problems when dealing with large-scale hypergraphs. The presence of a large number of nodes and hyperedges in hypergraphs makes the representation, storage, and computation of hypergraphs very complex. Especially in graph convolutional networks, processing large-scale hypergraphs may lead to a great consumption of computational and storage resources, thus affecting the efficiency of model training and inference. Therefore, future research could further improve the methodology to increase the accuracy and efficiency of the predictions. This method provides an innovative approach to thinking and a tool to study the association between miRNAs and diseases, which are conducive to gaining insight into the pathogenesis of diseases as well as developing new therapeutic approaches and applying them to more biomedical fields.

## Figures and Tables

**Figure 1 ncrna-10-00009-f001:**
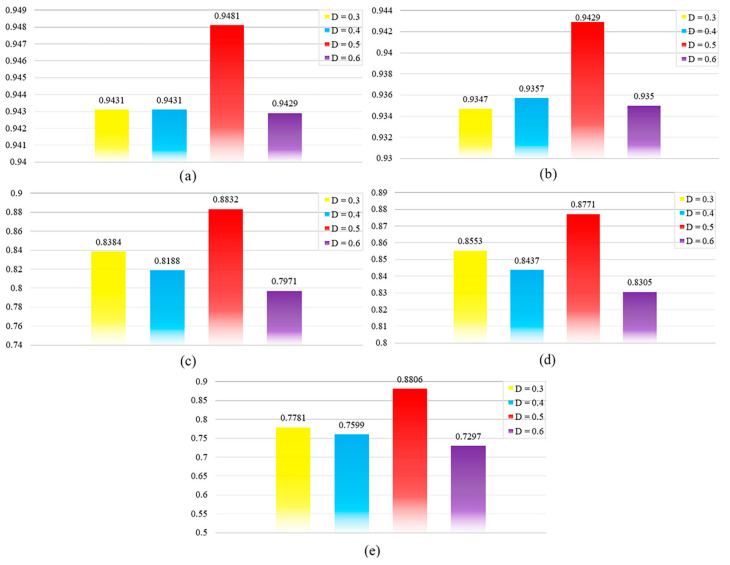
Parameter analysis of different dropouts. (**a**) AUC analysis for different dropout; (**b**) AUPRC analysis for different dropout; (**c**) Accuracy analysis for different drop-out; (**d**) F1-score analysis for different dropout; (**e**) Precision analysis for different dropout.

**Figure 2 ncrna-10-00009-f002:**
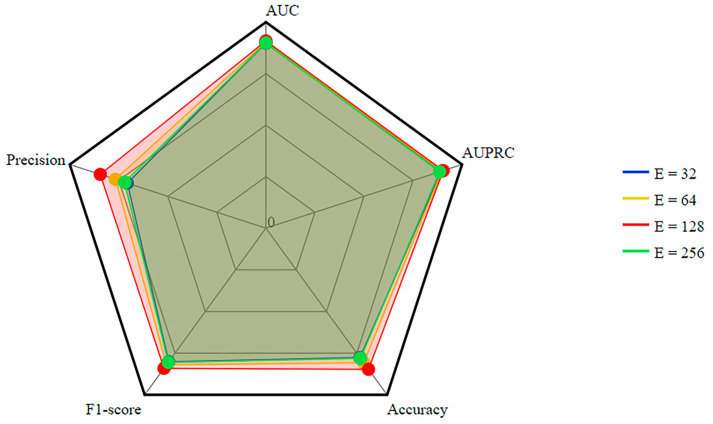
Parameter analysis for different feature embedding dimensions.

**Figure 3 ncrna-10-00009-f003:**
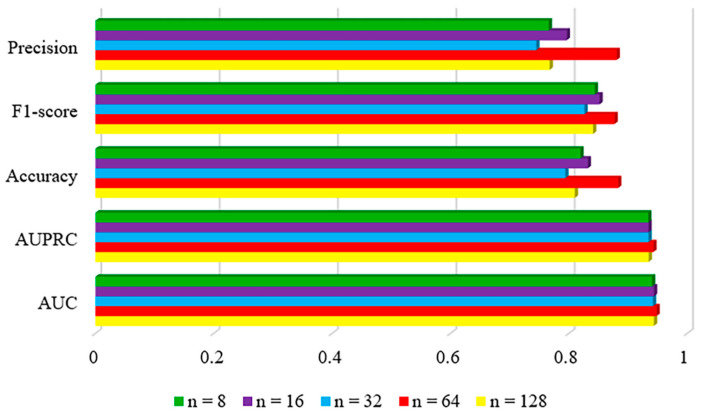
Parameter analysis for different numbers of hypernodes.

**Figure 4 ncrna-10-00009-f004:**
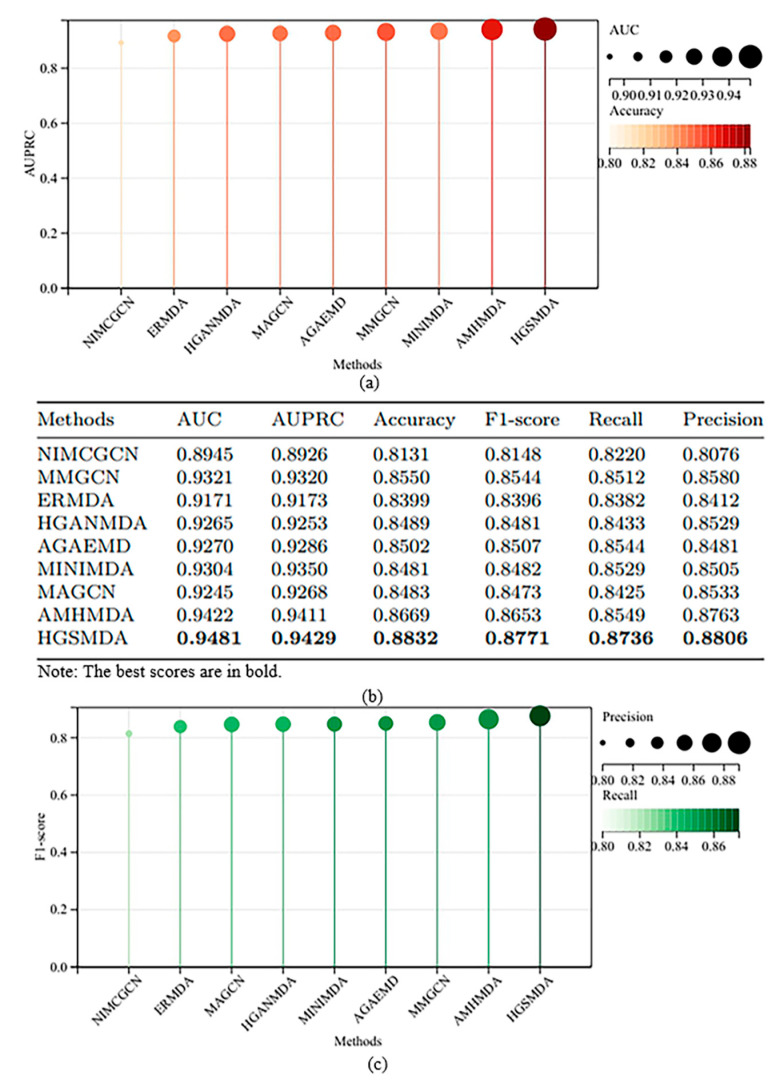
Comparison with other methods on HMDD v3.2. (**a**) AUC, AUPRC and Accuracy analysis for different method; (**b**) Detailed data on individual methods; (**c**) F1-score, Recall and Precision analysis for different method.

**Figure 5 ncrna-10-00009-f005:**
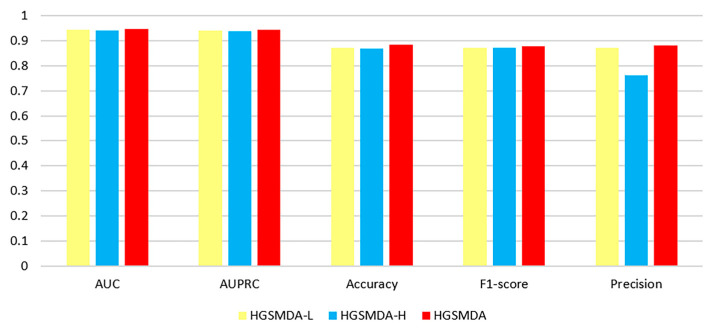
Results of ablation experiments for three different models.

**Figure 6 ncrna-10-00009-f006:**
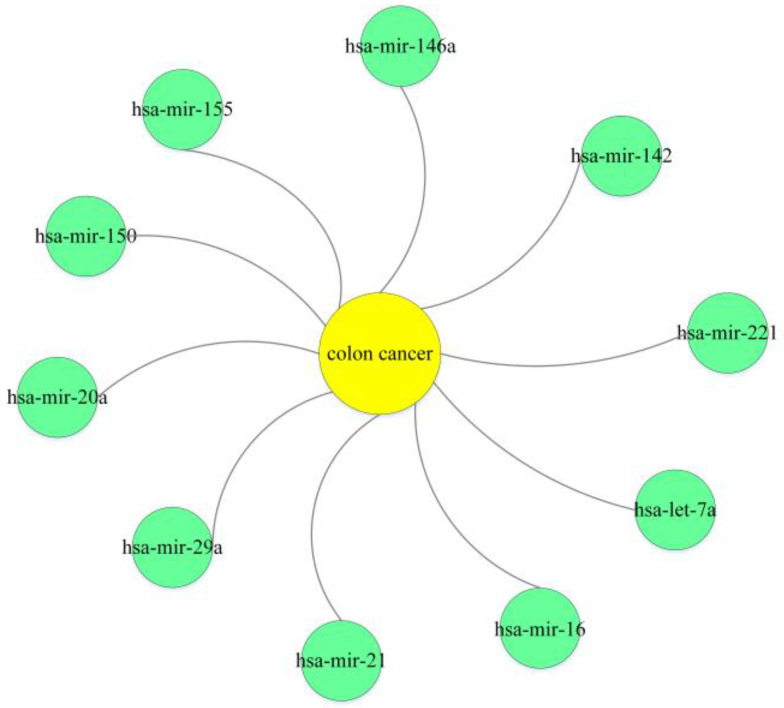
Prediction of the top 10 relevant miRNAs for predicting colon cancer.

**Figure 7 ncrna-10-00009-f007:**
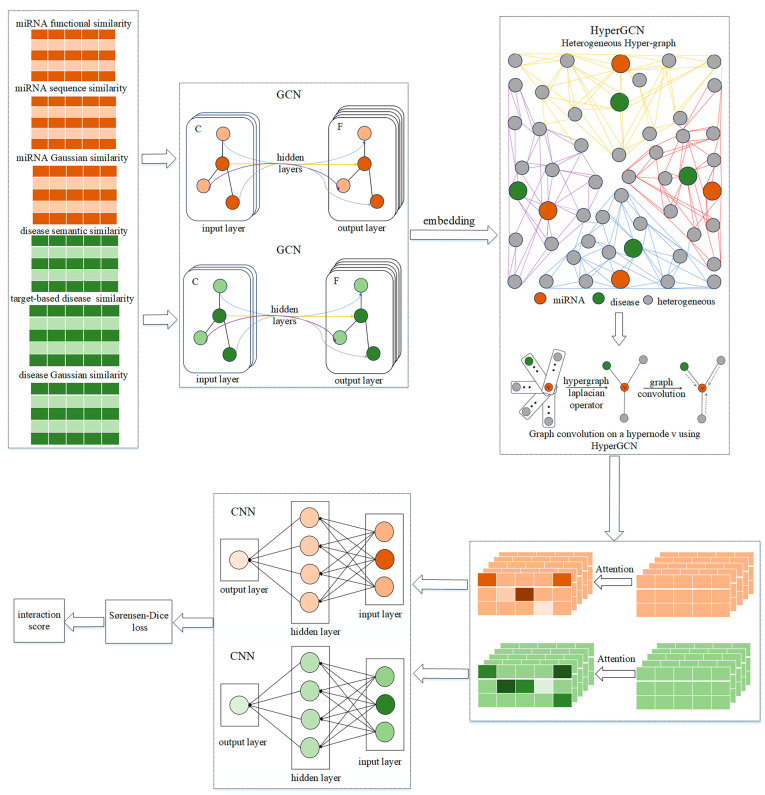
Architecture of the HGSMDA.

**Figure 8 ncrna-10-00009-f008:**
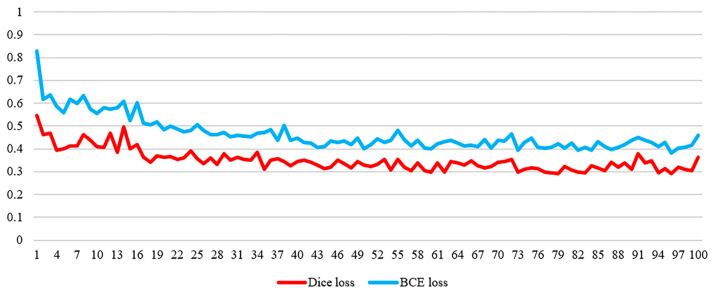
Loss values of the Sørensen-Dice loss function vs. the BCE loss function.

## Data Availability

This work utilizes data from publicaly available data set, the HMDD. The miRNA-diseasse database analyzed in the study is from HMDD v3.2, http://www.cuilab.cn/hmdd, accessed on 23 January 2024.

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
