# Peer review of "HGSMDA: miRNA–Disease Association Prediction Based on HyperGCN and Sørensen-Dice Loss"

_ncrna, 2024, doi:10.3390/ncrna10010009_

Round 1
Reviewer 1 Report
Comments and Suggestions for Authors The paper titled "HGSMDA: miRNA-disease association prediction based on HyperGCN and Sørensen-Dice loss" introduces a novel computational approach for predicting microRNA (miRNA) and disease associations. The study addresses the limitations of current Graph Convolutional Network (GCN) techniques in capturing complex relationships between miRNAs and diseases. The proposed method, HGSMDA, integrates HyperGCN and the Sørensen-Dice loss function to construct a miRNA-disease heteromorphic hypergraph, enabling more effective information aggregation. Experimental results using the Human MicroRNA Disease Database demonstrate that HGSMDA outperforms existing methods in prediction accuracy, providing a promising tool for exploring miRNA-disease associations and their role in disease pathogenesis. Strengths- Superior Performance Metrics: HGSMDA demonstrated outstanding performance in terms of accuracy and precision. Specifically, it achieved an AUC (Area Under Curve) of 94.81%, an AUPRC (Area Under the Precision-Recall Curve) of 94.29%, an accuracy of 88.32%, and a recall of 87.36%. This level of performance indicates a significant improvement over alternative approaches when evaluated on the Human MicroRNA Disease Database (HMDD) v3.2, a key resource for studying miRNA-disease associations.
- Innovative Loss Function: The Sørensen-Dice loss function used in HGSMDA is a major advancement over traditional loss functions like the BCE (Binary Cross Entropy) loss. This new loss function is stricter, taking into account both the intersection and concatenation of sets when calculating similarities. This results in more accurate predictions and a greater ability to minimize discrepancies between predicted and actual outcomes. The application of this function contributes to the model's enhanced predictive accuracy.
- Practical Validation and Application: The practical utility of HGSMDA is underscored by its successful application in a case study focused on colon cancer. This demonstrates the model's reliability and validity as a framework for exploring complex associations between miRNAs and diseases. Such practical validation shows that HGSMDA is not just theoretically sound but also applicable in real-world scenarios, making it a valuable tool for biomedical research.
- Enhanced Data Processing Techniques: For the challenge of calculating miRNA and disease similarity scores, the paper could benefit from integrating more advanced data preprocessing techniques. This could include the use of feature selection methods to identify the most relevant attributes or normalization techniques to standardize the data, leading to more accurate similarity scores.
- User-Friendly Software Implementation: Develop a user-friendly software tool or application based on the HGSMDA model. This tool should have a straightforward interface that allows clinicians and researchers to input data and receive predictions without needing in-depth technical knowledge. Providing clear guidance and documentation on how to use the tool would make it more accessible to a broader audience.
Cogent
Reviewer 2 Report
Comments and Suggestions for Authors
Dear Editors,
Dear Authors,
In the reviewed study the authors conducted a comprehensive study comparing their proposed model, HGSMDA, with eight other existing methods for miRNA-disease association prediction using the HMDD v3.2 dataset. The methods compared include NIMCGCN, MMGCN, ERMDA, HGANMDA, AGAEMD, MINIMDA, MAGCN, and AMHMDA. The initial similarity data, settings, parameters, and portions that dynamically acquire specific similarities for each method were maintained during the comparison. As a result, the manuscript presents HGSMDA as a robust model for miRNA-disease association prediction, showcasing superior performance compared to existing methods. The ablation experiments and case study further support the effectiveness of the proposed model, particularly highlighting the positive impact of the Sørensen-Dice loss function and the HyperGCN module.
The introduction provides a clear and concise overview of miRNAs and their crucial role in biological processes and diseases. It effectively establishes the motivation for studying miRNA-disease associations, emphasizing the limitations of traditional experiments and the emergence of computer-based methods as efficient alternatives. The introduction is structured logically, transitioning smoothly between different aspects of miRNA research and computational methods. Suggested improves:
1. Provide a brief explanation of the term "dichotomous network recommendation algorithm" to aid understanding;
2. Specify the nature of "prejudice evaluations" for clarity;
3. Clarify the term "inductive matrix completion approach" for better understanding;
4. Specify the characteristics that contribute to the "high quality of features" in machine learning models;
5. Specify the limitations or challenges associated with GCN in handling information from multiple nodes;
6. There is no mention about BLNIMDA – please refer also to this approach (Shang, J., Yang, Y., Li, F. et al. BLNIMDA: identifying miRNA-disease associations based on weighted bi-level network. BMC Genomics 23, 686 (2022). https://doi.org/10.1186/s12864-022-08908-8).
The information provided in the methodology as well as data and experiment chapters appear to be technically sound and relevant to the study. However, some sections could benefit from additional details or explanations:
1. Provide more context on the purpose and significance of constructing three different networks for miRNAs and diseases;
2. Elaborate on how the information is extracted using GCN, and how the initial embedding of GCN is randomized;
3. Explain the concept of hypernodes more explicitly and how they facilitate the creation of reliable connections;
4. Offer additional details on the neighborhood information update process, especially for readers less familiar with GCN;
5. Explain the rationale behind using the layer-level attention mechanism and how it contributes to the final node representations;
6. Provide a concise explanation of how the Sørensen-Dice loss function works and why it is chosen for this study.
The information provided in the Results and Discussion chapter appears to be technically sound. The comparisons between the proposed HGSMDA model and other methods are adequately presented, and the results of ablation experiments provide insights into the importance of different modules. The case study on colon cancer is relevant and adds practical context to the model's application. Suggested improvements:
1. The rationale behind using BCE Loss in HGSMDA-L and excluding HyperGCN in HGSMDA-H could be explained in more detail;
2. The explanation of HGSMDA-L and HGSMDA-H could be more detailed. A concise overview of the key changes introduced by each version would enhance clarity.
The conclusion chapter provides a clear summary of the proposed HGSMDA model and its application in predicting associations between miRNAs and diseases. The descriptions of the model's development and its evaluation methods are accurate. Suggestion:
1. Consider to provide additional context or examples to elucidate the challenges mentioned in calculating miRNA and disease similarity scores and handling large-scale hypergraphs.
The abstract is well-written, providing a clear overview of the study, the proposed methodology, and the experimental validation. It effectively conveys the problem, methodology, experimental validation, and implications of the proposed approach. No remarks.
The manuscript’s title is informative and provides a clear indication of the focus and methodology of the manuscript.
Overall, the manuscript is well-structured, and the methodology is novel, addressing limitations in conventional GCN techniques. The language is generally clear, but certain sections may benefit from minor revisions for improved clarity. The incorporation of additional depth in the literature review would enhance the manuscript's overall impact. With minor revisions, the manuscript has the potential to make a meaningful impact in the scientific community. Thus, I highly recommend to publish the reviewed manuscript in the ncRNA Journal after minor revision.
Best regards,
Comments on the Quality of English LanguageDear Editors,
Dear Authors,
In the reviewed study the authors conducted a comprehensive study comparing their proposed model, HGSMDA, with eight other existing methods for miRNA-disease association prediction using the HMDD v3.2 dataset. The methods compared include NIMCGCN, MMGCN, ERMDA, HGANMDA, AGAEMD, MINIMDA, MAGCN, and AMHMDA. The initial similarity data, settings, parameters, and portions that dynamically acquire specific similarities for each method were maintained during the comparison. As a result, the manuscript presents HGSMDA as a robust model for miRNA-disease association prediction, showcasing superior performance compared to existing methods. The ablation experiments and case study further support the effectiveness of the proposed model, particularly highlighting the positive impact of the Sørensen-Dice loss function and the HyperGCN module.
The introduction provides a clear and concise overview of miRNAs and their crucial role in biological processes and diseases. It effectively establishes the motivation for studying miRNA-disease associations, emphasizing the limitations of traditional experiments and the emergence of computer-based methods as efficient alternatives. The introduction is structured logically, transitioning smoothly between different aspects of miRNA research and computational methods. Suggested improves:
1. Provide a brief explanation of the term "dichotomous network recommendation algorithm" to aid understanding;
2. Specify the nature of "prejudice evaluations" for clarity;
3. Clarify the term "inductive matrix completion approach" for better understanding;
4. Specify the characteristics that contribute to the "high quality of features" in machine learning models;
5. Specify the limitations or challenges associated with GCN in handling information from multiple nodes;
6. There is no mention about BLNIMDA – please refer also to this approach (Shang, J., Yang, Y., Li, F. et al. BLNIMDA: identifying miRNA-disease associations based on weighted bi-level network. BMC Genomics 23, 686 (2022). https://doi.org/10.1186/s12864-022-08908-8).
The information provided in the methodology as well as data and experiment chapters appear to be technically sound and relevant to the study. However, some sections could benefit from additional details or explanations:
1. Provide more context on the purpose and significance of constructing three different networks for miRNAs and diseases;
2. Elaborate on how the information is extracted using GCN, and how the initial embedding of GCN is randomized;
3. Explain the concept of hypernodes more explicitly and how they facilitate the creation of reliable connections;
4. Offer additional details on the neighborhood information update process, especially for readers less familiar with GCN;
5. Explain the rationale behind using the layer-level attention mechanism and how it contributes to the final node representations;
6. Provide a concise explanation of how the Sørensen-Dice loss function works and why it is chosen for this study.
The information provided in the Results and Discussion chapter appears to be technically sound. The comparisons between the proposed HGSMDA model and other methods are adequately presented, and the results of ablation experiments provide insights into the importance of different modules. The case study on colon cancer is relevant and adds practical context to the model's application. Suggested improvements:
1. The rationale behind using BCE Loss in HGSMDA-L and excluding HyperGCN in HGSMDA-H could be explained in more detail;
2. The explanation of HGSMDA-L and HGSMDA-H could be more detailed. A concise overview of the key changes introduced by each version would enhance clarity.
The conclusion chapter provides a clear summary of the proposed HGSMDA model and its application in predicting associations between miRNAs and diseases. The descriptions of the model's development and its evaluation methods are accurate. Suggestion:
1. Consider to provide additional context or examples to elucidate the challenges mentioned in calculating miRNA and disease similarity scores and handling large-scale hypergraphs.
The abstract is well-written, providing a clear overview of the study, the proposed methodology, and the experimental validation. It effectively conveys the problem, methodology, experimental validation, and implications of the proposed approach. No remarks.
The manuscript’s title is informative and provides a clear indication of the focus and methodology of the manuscript.
Overall, the manuscript is well-structured, and the methodology is novel, addressing limitations in conventional GCN techniques. The language is generally clear, but certain sections may benefit from minor revisions for improved clarity. The incorporation of additional depth in the literature review would enhance the manuscript's overall impact. With minor revisions, the manuscript has the potential to make a meaningful impact in the scientific community. Thus, I highly recommend to publish the reviewed manuscript in the ncRNA Journal after minor revision.
Best regards,
